# Systematic Review of the Genus Nalepa Reitter, 1887 (Coleoptera, Tenebrionidae, Blaptinae, Blaptini) from the Tibetan Plateau, with Description of Six New Species and Two Larvae [note 1]

**DOI:** 10.3390/insects13070598

**Published:** 2022-06-29

**Authors:** Xiu-Min Li, Juan Tian, Jiao-Jiao Fan, Guo-Dong Ren

**Affiliations:** Key Laboratory of Zoological Systematics and Application of Hebei Province, College of Life Sciences, Institute of Life Science and Green Development, Hebei University, Baoding 071002, China; lixiumin@hbu.edu.cn (X.-M.L.); juantian521@163.com (J.T.); fjj15635452262@163.com (J.-J.F.)

**Keywords:** *Nalepa*, molecular delimitation, morphology, new species, larva

## Abstract

**Simple Summary:**

The Tibetan Plateau is the largest and highest plateau in the world. The Tibetan Plateau is widely considered as a large natural experimental area for speciation; its uplift has facilitated allopatric speciation and diversification since the Miocene. In general, the Tibetan Plateau is known for its large number of endemic species. In this study, we revise the systematics of the endemic Tibetan genus *Nalepa* Reitter, 1887 (Blaptini tribe) and describe six new species based on larval and adult morphology and molecular data. We also provide a diagnostic key to the members of the genus *Nalepa*. Larvae were associated with the adults using a combined molecular species delimitation, and adult males and females are described and illustrated. Phylogenetic relationships of the members of the genus *Nalepa* are provided and discussed based on fragments of the mitochondrial and nuclear markers. Further, we applied molecular species delimitation methods to verify the taxonomic status of the new species. Lastly, the status of the genus *Nalepa* from the Tibetan Plateau is briefly discussed.

**Abstract:**

*Nalepa* Reitter, 1887 is a monotypic genus of the tenebrionid tribe Blaptini and is endemic to the Tibetan Plateau. In this study, the genus *Nalepa* was reviewed using a combination of molecular and morphological datasets. Based on the results, six new species were described: *N.*
*acuminata*
**sp. n.**, *N. ovalifolia*
**sp. n.**, *N.*
*polita*
**sp. n.**, *N. quadrata*
**sp. n.,**
*N.*
*xinlongensis*
**sp. n.,** and *N.*
*yushuensis*
**sp. n.** In addition, the larvae of *N. cylindracea* Reitter, 1887 and *N. quadrata*
**sp. n.** were described and associated with related adults using molecular approaches. This study provides valuable molecular and morphological data for phylogenetic studies.

## 1. Introduction

The subfamily Blaptinae was resurrected based on molecular phylogenetic analyses and contains seven tribes: Amphidorini, Blaptini, Dendarini, Pedinini, Platynotini, Platyscelidini, and Opatrini [1]. The tribe Blaptini was currently divided into five subtribes and 28 genera (about 500 species), and nine of these genera contain only one species, including the genus *Nalepa* Reitter, 1887 [2,3,4,5]. *Nalepa* was erected by Reitter in 1887 as a subgenus of the genus *Blaps*: *Blaps* (*Nalepa*) Reitter, 1887 (type species: *Blaps cylindracea* Reitter, 1887; type locality: Amdo). Reitter revised the status of *Nalepa* and elevated it to a genus in 1893. Later, Ren et al. reviewed the tribe Blaptini from China in 2016, and proposed a new synonymy: *Nalepa cylindracea* Reitter, 1887 = *Blaps ratalaria* Ren Wang, et al., 2001 (collected from Tangpu, Jomda, Xizang, China) [5,6]. To date, the genus *Nalepa* Reitter, 1887 includes one species that is flightless and well-adapted to semi-arid and arid environments. It is an endemic genus of the Tibetan Plateau distributed in Amdo (probably located in Qinghai at present), Jomda of Xizang, and Zadoi of Qinghai.

Larval morphology is important for understanding the systematics of different groups of Coleoptera, and it has been used to support the close relationships of genera or subtribes [7,8,9,10,11]. In the subfamily Blaptinae, the available data on larval morphology are mostly from the tribe Opatrini [12]. For the description of larval morphology within Blaptini, Yu et al. made valuable contributions and gave keys to the larvae of common species for this group [13,14,15,16,17]; however, the characteristics of larva for the genus *Nalepa* is still unknown due to the lack of specimens.

In this study, six new species, the larvae of *Nalepa cylindracea* Reitter, 1887, and a new species are described based on morphological and molecular evidence. In addition, we construct a molecular phylogeny for the genus and apply molecular species delimitation methods to verify the taxonomic status of the new species. 

## 2. Material and Methods

### 2.1. Morphological Examination

In total, 701 specimens (including larvae) from 47 sampling localities were examined for this study and have been deposited at the Museum of Hebei University, Baoding, China (MHBU). The figures were taken with three distinct imaging systems: (a) Canon EOS 5D Mark III (Canon Inc., Tokyo, Japan) connected to a Laowa FF 100 mm F2.8 CA-Dreamer Macro 2× or Laowa FF 25 mm F2.8 Ultra Macro 2.5–5× (Anhui Changgeng Optics Technology Co., Ltd., Hefei, China); (b) Leica M205A stereomicroscope equipped with a Leica DFC450 camera (Leica Microsystems, Singapore, Singapore), which was controlled using the Leica application suite 4.3; (c) JVC^®^ KY-F75U (JVC Kenwood, Long Beach, CA, USA) digital camera attached to a Leica Z16 APO dissecting microscope (Leica Microsystems, Buffalo Grove, IL, USA) with an apochromatic zoom objective and motor focus drive, using a Syncroscopy^®^ Auto-Montage System (Synoptics, Cambridge, UK) and software. Multiple images were used to construct the final figures. Images were illuminated with either an LED ring light attached to the end of the microscope column, with incidental light filtered to reduce glare, or by a gooseneck illuminator with bifurcating fiberoptics; image stacks were white balance corrected using the system software (Synoptics, Cambridge, UK).

Label data are presented verbatim. A double slash (//) separates text on different lines of label; authors’ remarks are enclosed in brackets “[]”. 

### 2.2. Taxon Sampling, DNA Extraction, PCR Amplification, and Sequencing

Larval specimens were collected in the field together with adults from the Tibetan Plateau. In an attempt to associate the different stages, the molecular data were collected for 29 individuals (27 adults and 2 larvae). Detailed information for all the samples used in this study (including those without molecular data) is provided in Appendix A. 

DNA was extracted from head tissue of larva and leg muscle tissue of adults using Insect DNA isolation Kit (BIOMIGA, Dalian, China), following the manufacturer’s protocols. The DNA extracted was stored at −20 °C. Fragments of four mitochondrial markers (cytochrome coxidase subunit I, COI (two fragments), cytochrome coxidase subunit Ⅱ, COⅡ; cytochrome b, Cytb; 16S ribosomal DNA, 16S), and one nuclear marker (28S ribosomal DNA domain D2, 28S-D2) were amplified and sequenced. The primers and annealing temperatures are shown in Table 1.

The profile of the PCR amplification consisted of an initial denaturation step at 94 °C for 4 min, 35 cycles of denaturation at 94 °C for 1 min, annealing for 45 s, an extension at 72 °C for 1 min, and a final 8 min extension step at 72 °C. PCR was performed using TaKaRa Ex Taq (TaKaRa, Dalian, China). PCR products were subsequently checked by 1% agarose gel electrophoresis and sequencing was performed at GENERAL BIOL Co., Ltd. (Anhui, China). Altogether, 144 new sequences from 29 individuals of six species were generated. We used previously published sequences of *B. rhynchoptera* Fairmaire, 1986 as the outgroup (Accession number: MK854717), which has been recovered as a close relative of *Nalepa* [21]. 

### 2.3. Phylogenetic Analyses and Molecular Species Delimitation

Phylogenetic analyses were carried out on the concatenated dataset under maximum likelihood (ML) criterion. For ML analyses, we used IQ-TREE v1.6.6 [22] as implemented in the dedicated IQ-TREE web server (http://iqtree.cibiv.univie.ac.at/, accessed on 1 June 2022). The ML tree was inferred under an edge-linked partition model for 5000 ultrafast bootstraps [23]. The consensus phylogenetic tree was visualized in Figtree v.1.4.4. (http://tree.bio.ed.ac.uk/software/figtree, accessed on 1 June 2022).

For the molecular species delimitation analyses, we applied a combination of five distinct methods/settings to assess the boundaries of species within *Nalepa*. The automatic barcode gap discovery (ABGD) approach as implemented on the online web application (https://bioinfo.mnhn.fr/abi/public/abgd/abgdweb.html, accessed on 5 May 2022) was carried out on the *COI*, *COII*, and *Cytb* genes, but not on the whole multi-locus dataset in order to avoid biases in the estimations of distance associated with missing data; the outgroup was also excluded. We used the Kimura 2-parameter distance and the following settings: the range of prior intraspecific divergences (*p*) from 0.001 to 0.1; 10 recursive steps; minimum relative gap width of 0.5; the number of bins of 20 for distance distribution. To determine the number of putative species, we favored priors between 0.01 and 0.03, as they tend to be more conservative [24]. In addition to the distance-based ABGD method, we also performed tree-based analyses using two distinct methods: the general mixed Yule coalescent (GMYC) model and Poisson-tree-processes (PTP) analyses of molecular species delimitation [25,26]. The first one relied on the best-score ML tree from the IQ-TREE analysis on the concatenated dataset, with the outgroup removed. Then, the resulting trees were further used for GMYC and PTP analyses. PTP analyses were carried out on the web server of the Exelixis Lab (http://species.h-its.org/ptp/, accessed on 6 May 2022), using the default settings. 

## 3. Results

### 3.1. Morphological Study and Diagnosis

Body long cylindrical, medium to large (17–22 mm), black and shiny. Antennae moderately long, often reaching pronotum base when directed backwards (male), antennomere VII equal to or slightly longer than antennomeres V-Ⅵ, antennomeres Ⅷ-Ⅴ spherical, Ⅺ spindle-shaped. Disc of pronotum convex, not flattened, subquadrangular or trapezoidal (base wider than anterior margin). Elytra subcylindrical, apex thick and obtuse with fine punctuation, without mucro. Legs long, profemur strong, metatarsomere Ⅰ longer than others and cylindrical.


**Key to species of the genus *Nalepa* Reitter, 1887 (based on males)**
Parameres bottleneck-shaped...........................................................................................2
-Parameres cone-shaped...........................................................................................3
2.Pronotum trapezoid-shaped, wide at base; arcuately narrowing from basal 1/3 to apex laterally in dorsal view; antennomer Ⅶ longer than Ⅴ and Ⅵ one...........................................................................................*N. acuminata*
**sp. n.** Li & Ren, **sp. n.**
-Pronotum square-shaped, sides parallel; arcuately narrowing from basal 1/3 to apex laterally in dorsal view; antennomer Ⅴ-Ⅶ nearly same length...........................................................................................*N. xinlongensis* Li & Ren, **sp. n.**
3.Parameres cone-shaped, straightly narrowing from basal to apex (Figure 1C,D).....4
-Parameres lateral margin arcuately narrowing from base to 1/3, straight narrowing from basal 1/3 to apex...........................................................................................*N. yushuensis* Li & Ren, **sp. n.**
4.Pronotum and elytra strongly convex on disc, lateral margins of pronotum subparallel; male elytra more oval, surface smooth and shiny...*N. cylindracea* Reitter, 1887
-Pronotum and elytra weakly convex on disc, lateral margins of pronotum nearly parallel; male elytra long ovoid....................................................*N. ovalifolia* Li & Ren **sp. n.**
5.Antennomer VII shorter than V one; parameres straightly narrowing, apex obtuse (Figure 3C,K).............................................................................................. *N.*
*quadrata* Li & Ren **sp. n.**
-antennomer VII longer than V one; parameres straightly narrowing, apex acuminate (Figure 6C,K)................................................................ *N.*
*polita* Li & Ren **sp. n.**



***Nalepa cylindracea* Reitter, 1887 (Figure 1A,B)**


*Nalepa cylindracea* Reitter, 1887: 366; Löbl et al., 2008: 229. Type locality: China, Amdo.

*Blaps cylindracea*: Reitter, 1893b: 316. 

*Blaps ratalaria* Ren Wang, et al., 2001: 22–23. Synonymized by Ren et al., 2016: 198.

**Adult**. 

Male (Figure 1A). Body length 17.0–20.0 mm, weakly shiny, oval-oblong. Head widest at eye level, with coarsely punctate; anterior margin of epistome slightly emarginated; antennae long and slender, reaching pronotal base when directed backwards, antennomeres VIII-X spherical, XI spindle-shaped. Pronotum long and wide nearly equal, square, lateral margins from middle narrowing toward anterior angles arcuated, external rim complete. Disc strongly convex, smooth, surface with dense and fine punctation. Elytra long ovoid, widest after the middle, weakly arcuated; 2.5 times as long as pronotum; Elytral surface smooth, with sparse punctures at base, apex of elytra steeply sloping, obtuse. Legs long, profemur strong, protibiae with internal and external nearly straight; protarsomeres I cylindrical, extension toward apex. 

Female (Figure 1B). Body wider than male. Antennae shorter than male. Pronotum square. Elytra more convex, 1/3 backward extension.

**Male genitalia** (Figure 1C–E) Aedeagus length 4.4–4.7 mm and width 1.0–1.1 mm. Parameres length 1.4–1.5 mm and width 0.7–0.8 mm; parameres widest and convex at base, cone-shaped, straightly narrowing from base to apex laterally in dorsal view, apex acuminate; dorsal side nearly straight, slightly curved to ventral side apically in lateral view; parameres narrowed almost in a straight line up to apex in lateral view, apex weakly bend.


**Larva.**


**Material examined.** Larvae were collected in the field together with the following collecting information: 2 exx. in ethanol, “2018.Ⅷ.21//Ya’gando, Baqêeg, Xizang, China//Xinglong Bai et al.//31°55.916′ N//96°28.051′ E//Elev. 4432 m//Museum of Hebei University”; 1 exx. in ethanol, “2012. VII. 22//Saiqu, Zadoi, Qinghai, China//Guodong Ren et al.//32°50.783′ N//95°28.634′ E//Elev. 4045 m//Museum of Hebei University”. 

**Body** (Figure 2A–C)**.** Length 30.0–35.0 mm. Body subcylindrical with sharp tail-end; yellowish-brown; vestiture smooth; median line obvious on first five segments; covered with four pairs long setae on each tergite (besides the last segment); evenly sclerotized dorsally and ventrally; posterior border of each segment dark brown, with longitudinal parallel stripes. 

**Head** (Figure 2D,E): Prognathous, slightly narrower than the width of prothorax, slightly convex dorsally; sides rounded (Figure 2D). Labrum transverse, weakly emarginate, apical margin with two rows of setae. Mandibles well developed, elongate, left and right sides asymmetrical, with two pairs of short setae; elongate anterior extension. Clypeus transverse, trapezoid-shaped, feebly convex, anterior margin slightly linear, and with four long erect clypeal setae. Epicranial stem Y-shaped (Figure 2E); frons and epicranial plate slightly convex, with dense and fine punctures and wrinkles form irregularly polygonal, center of anterior rim of frons with one pair of hairs. Maxillary palps 3-segmented, segments I and II subcylindrical, III fingerlike; I widest, II and III with the same length, twice as long as I (Figure 2D,E). Labial palps (Figure 2D) 2-segmented, short, II fingerlike. Mentum oval-trapezoidal, base of mentum straight, prementum with two long hairs. Antennae well developed, three-segmented dome-like, cylindrical, shorter than length of head, segment I noticeably wider; II and III similar in shape, about twice as wide as I.

**Thorax** (Figure 2A–C): Thoracic segmentation C-shaped in dorsal view, sides parallel, widest in the middle, with transverse plicae. Each thoracic tergum with long slender setae on the sides of anterior and posterior margins. Pronotum longest, about 0.8 times as long as the sum of meso- and metanotum; mesonotum shortest.

**Abdomen** (Figure 2F,G): Approximately 3.8 times as long as thorax; segments I-VIII subcylindrical with transverses plicae, faintly rugose, with sparse elongate setae ventrally. Tergum IX is nearly the same length and distinctly narrower than VIII, marginal with row of short spines (six spines on the left side; seven spines on the right side), pygopods subconical in dorsal view; urogomphi suddenly upturned to apex in lateral view, with three short spines at about middle; surface of the convex disc with sparse long setae in ventral view. 

**Legs** (Figure 2B–E): Legs are well developed. Prothoracic leg is noticeably longer, much thicker than meso- and metathoracic ones; profemur and protibia with a row of spines and denser long setae (Figure 2E). Protarsungulus strongly sclerotized sharp claw-like, profemora and protibiae gradually narrowing towards apex; profemora about 0.8 times length of protibia; meso- and metathoracic legs moderately shorter than prothoracic one, tarsungulus highly ossified hook-like, with a row of spines and sparse setae.

**Spiracles** (Figure 2B): a pair of well-developed, round thoracic spiracles, situated ventrolaterally on anterolateral margins of terga I-VIII.


***Nalepa**quadrata* Li & Ren sp. n. (Figure 4A,B)**


**Type locality**. China, Sichuan Province, Luhuo County, Simu.

**Type specimens** (Adults). **Holotype:** ♂, with the following labels: “2016.VIII.3//Simu Township, Luhuo County, Sichuan, China//Xiu-Min Li et al.//Museum of Hebei University”//“31°19.813′ N//100°44.088′ E//Elev. 3180 m//Museum of Hebei University”. **Paratypes:** 34♂14♀ [1♀ in ethanol] (MHBU), same data as holotype; 1♂2♀ in ethanol (MHBU), labeled “2021.VII.17//Luhuo County G350 road, Garzê, Sichuan, China//Xiu-Min Li et al.//Museum of Hebei University]”//“31°32.867′ N//100°73.436′ E//Elev. 3094 m//Museum of Hebei University]”; 14♂6♀ [1♂1♀ in ethanol] (MHBU), labeled “2016. VIII.14//Lean Township, Xinlong County, Garzê, Sichuan, China//Xiu-Min Li et al.//Museum of Hebei University”//“31°12.649′ N//100°17.969′ E//Elev. 4015 m//Museum of Hebei University”.

**Other material. Larvae.** 2 exx. In ethanol “2016.VIII.3//Simu Township, Luhuo County, Sichuan, China//Xiu-Min Li & Xing-Long Bai et al.//Museum of Hebei University”//“31°19.813′ N//100°44.088′ E//Elev. 3180 m//Museum of Hebei University”.

Body length 18.0–18.3 mm, width 7.5–7.6 mm; black, weakly shiny, oval-oblong. 

**Head** (Figure 3A,B). Anterior margin of epistome emarginated. Lateral margins of epistome straight. Lateral margins of the head with indistinct emargination between epistome and genae. Head widest at eye level. Mentum transverse, with elliptical lateral sides, coarsely punctate and slightly impressed in the middle of the anterior edge. Antennae long and slender, reaching pronotal base when directed backwards, antennomeres VIII-X spherical, XI spindle-shaped. Ratio of length/width of antennomeres II-XI, 6.0(8.0):21.0(8.0):10.0(8.0):10.0(8.0):11.0(8.0):10.0(8.0):8.0(9.0):8.0(9.0):8.0(9.0):14.0(8.0).

**Thorax** (Figure 3D). Transverse, 1.6 times as wide as head. Broadest in the middle, lateral margins weakly sinuous before posterior angles toward the middle, nearly subparallel, then toward anterior angles arcuately narrowed, ratio of width at the anterior margin to the middle part and base 14:23:22, external rim complete. Disc convex, smooth, surface with dense punctation. Anterior angles are obtuse; posterior ones are weakly obtuse. Prosternal process obliquely sloping behind procoxae, distinctly projecting beyond the margin of prosternum.

**Abdomen.** Long ovoid, broadest around in the middle, strongly convex on disc, 1.6 times as long as wide, widest at the middle, weakly arcuated; 2.3 times as long and 1.2 times as wide as pronotum; 1.9 times as wide as head. Elytral surface smooth, with sparse punctures and dense irregular short wrinkles, apex of elytra steeply sloping, obtuse. Abdomen without hair tuft/short yellow setae between 1st and 2nd abdominal ventrites, 1st—3rd with transverse/longitudinal wrinkles, abdominal ventrites 4th—5th with dense punctures and simple particles.

**Legs** (Figure 3E–I). Legs long, profemora strong, protibiae nearly straight. Ventral surface of protarsomeres I-IV with a hairy brush. Inner side of the mesotibiae weakly curved, apex extension; ventral surface of the mesotarsomeres I–IV with a hairy brush. Ratio of length (width) of pro-, meso-, and metatibiae 28.0(4.0):35.0(4.0):41.0(4.0), that of metatarsomeres I-Ⅳ, 13.0(4.0):7.0(4.0):7.0(4.0):14.0(5.0).

**Male genitalia** (Figure 3K–M) 2.8 mm long and 0.8 mm wide. Parameres 1.0 mm long and 0.7 mm wide, conical shape; parameres wide at the base, straight narrowing in dorsal view, apex obtuse; rods of gastral spicula merged at the apex (Figure 3K). Weakly curved, parameres narrowed almost in a straight line up to the apex in lateral view.

**Sexual dimorphism** (Figure 4B). Body length 18.9–19.1 mm, width 8.9–9.0 mm, wider than male. Head 1.3 times as wide as interocular distance, pronotum 1.3 times as wide as long, elytra 1.4 times as long as wide. Antennae shorter than male, not reaching pronotal base; antennomeres VIII-X spherical. Pronotum widest in the middle, lateral margins nearly subparallel from base to 2/3, then straight narrowing toward anterior angles. Elytra is more convex. The inner side of the protibiae is nearly straight, the upper spur sharper apically, almost to the end of protarsomere I. Ventral surface of pro- and mesotarsomeres with a hairy brush.


**Larva.**


**Body** (Figure 5A–C). Length 26.1–28.0 mm. Body subcylindrical with sharp tail-end; yellowish-brown; vestiture smooth; Median line is obvious at the first five segments; covered with four pairs of long setae on each tergite (besides the last segment); evenly sclerotized dorsally and ventrally; posterior border of each segment dark brown, with longitudinal parallel stripes. 

**Head** (Figure 5D,E). Prognathous, slightly narrower than the width of prothorax, slightly convex dorsally, sides rounded (Figure 5D); Labrum transverse, weakly emarginate, apical margin with two rows of setae. Mandibles well developed, elongate, left and right sides symmetrical, with two pairs of short setae, elongate anterior extension. Clypeus transverse, trapezoid-shaped, feebly convex, anterior margin slightly linear, and with four long erect clypeal setae. Epicranial stem Y-shaped (Figure 5E); frons and epicranial plate slightly convex, with dense and fine punctures and wrinkles form irregularly polygonal, the center of anterior rim of frons with one pair of hairs (Figure 5E). Maxillary palps are three-segmented, subcylindrical, with cone-shaped at peak; I widest, II and III with the same length, twice as long as I (Figure 5D,E). Labial palps are two-segmented, short, II cone-shaped (Figure 5D). Mentum oval-trapezoidal, base of mentum straight, prementum with two long hairs. Antennae well developed, three-segmented dome-like, cylindrical, approximate cylindrical, shorter than the length of head, segment I noticeably wider; III is longest, about 1.5 times as long as the II, and 0.8 times as wide as II. 

**Thorax** (Figure 5A–C): Thoracic segmentation C-shaped in dorsal view, sides parallel, widest in the middle, with transverses plicae; Each thoracic tergum with long slender setae on sides of anterior and posterior margins; Pronotum longest, about 0.8 times as long as the sum of meso- and metanotum; mesonotum shortest.

**Abdomen** (Figure 5A–C,F,G): Approximately 3.8 times as long as thorax; segments I-VIII subcylindrical with transverses plicae, faintly rugose, with sparse elongate setae ventrally. Tergum IX distinctly narrower than VIII, pygopods subconical in dorsal view, margin with row of short spines (seven spines on the left side; six spines on the right side), urogomphi with three short spines at middle; urogomphi suddenly upturned to apex in lateral view; disc with sparse long setae in ventral view (Figure 5F,G). 

**Legs** (Figure 5B–D): Legs well developed. Prothoracic leg noticeably longer, much thicker than meso- and metathoracic ones. Profemur with a close row of spines and denser long setae (Figure 5E). Protarsungulus strongly sclerotized sharp claw-like, profemur and protibia gradually narrowing towards apex, profemora about 0.8 times length of protibia; meso- and metathoracic legs moderately shorter than prothoracic one, tarsungulus highly ossified hook-like, with a row of spines and denser long setae.

**Spiracles** (Figure 5B). A pair of well-developed, round thoracic spiracles, situated ventrolaterally on anterolateral margins of terga I–VIII.

**Etymology.** This species is named from the Latin adjective “*quadratus*, *-a*, *um*”, in reference to its pronotum nearly square.

**Distribution.** China, Sichuan.

**Diagnosis.** The adult of the new species is morphologically very similar to *N. polita* Li & Ren **sp. n.**, but can be distinguished from the latter by the following male characters: (1) pronotum and elytra strongly convex on disc (pronotum and elytra convex in *N. polita*); (2) antennomer Ⅶ shorter than Ⅴ one (antennomer Ⅶ longer than Ⅴ one in *N. polita*); (3) parameres apex obtuse (parameres apex acuminate n *N. polita*)

The larva of the new species is morphologically very similar to *N. cylindracea*, but can be distinguished from the latter by the following male characters: (1) profemur with a close row of spines (with irregular arranged few spines in *N. cylindracea*); (2) marginal of tergum IX with a row of short spines: seven spines on the left side, six spines on the right side (marginal of tergum IX with a row of short spines: left side six spines, right side seven spines in *N. cylindracea*); (3) Antennae III is longest, about 1.5 times as long as the II, and 0.8 times as wide as II (Antennae II and III similar in shape in *N. cylindracea*).


***Nalepa polita* Li & Ren sp. n. (Figure 8A–D)**


**Type locality**. China, Sichuan Province, Dêgê County, Chowa.

**Type specimens** (Adults). Holotype: ♂, with the following labels: “2011.VII.26//Chowa Township, Dêgê County, Garzê, Sichuan, China//Xiu-Min Li et al. //Museum of Hebei University”. **Paratypes:** 29♂6♀(MHBU), same data as holotype; 13♂5♀(MHBU), labeled “2009.VII.18//Yiniu Township, Shiqu County, Sichuan, China//Xiu-Min Li et al. //Museum of Hebei University”//“33°0361′ N//98°3176′ E//Elev. 3937 m//Museum of Hebei University”; 1♂1♀(MHBU), labeled “2009.VII.17//Sêrxü, Township, Shiqu County, Sichuan, China//Xiu-Min Li *et al*.//Museum of Hebei University”//“33°1346′ N//97°8376′ E//Elev. 4113 m//Museum of Hebei University”; 2♂2♀ [2♂ in ethanol] (MHBU), labeled “2016.VIII.5//Gyawa Township, Garzê County, Sichuan, China//Xiu-Min Li et al. //Museum of Hebei University”//“31°34.055′ N//99°58.205′ E//Elev. 3670 m//Museum of Hebei University”; 2♂ [2♂ in ethanol] (MHBU), labeled “2016. VIII.3//Rinda Township, Luhuo County, Garzê, Sichuan, China]//Xiu-Min Li et al.//Museum of Hebei University]”//“31°13.082′ N//100°50.421′ E//Elev. 3092 m//Museum of Hebei University”.

**Description.** Body length 17.8–18.0 mm, width 7.9–8.1 mm; black, weakly shiny, oval-oblong. 

**Head** (Figure 6 and Figure 7A,B). Anterior margin of epistome emarginated. Lateral margins of epistome straight. Lateral margins of head with indistinct emargination between epistome and genae. Head widest at eye level. Mentum transverse, with elliptical lateral sides, coarsely punctate and slightly impressed in the middle of the anterior edge. Antennae long and slender, reaching pronotal base when directed backwards, antennomeres VIII-X spherical, XI spindle-shaped. Ratio of length/width of antennomeres II-XI 6.0(8.0):26.0(8.0):12.0(8.0):12.0(8.0):13.0(7.0):13.0(8.0):10.0(9.0) 10.0(9.0):10.0(9.0):13.0(9.0).

**Thorax** (Figure 6 and Figure 7C). Transverse, 1.6 times as wide as head. Widest in the middle, lateral margins weakly broading toward the middle, then toward anterior angles arcuated, ratio of width at the anterior margin to the middle part and base 25:40:36, external rim complete. Disc convex, smooth, surface with dense punctation. Anterior angles obtuse, posterior ones weakly obtuse. Prosternal process obliquely sloping behind procoxae, distinctly projecting beyond the margin of prosternum.

**Abdomen.** Long ovoid, 1.6 times as long as wide, widest in the middle, weakly arcuated, 2.4 times as long and 1.5 times as wide as pronotum, 2.0 times as wide as head. Strongly convex on disc, elytral surface smooth, with sparse punctures and dense irregular short wrinkles, apex of elytra steeply sloping, obtuse. Abdomen without hair tuft/short yellow setae between 1st and 2nd abdominal ventrites, 1st—3rd with transverse/longitudinal wrinkles, abdominal ventrites 4th—5th with dense punctures and simple particles.

**Legs** (Figure 6 and Figure 7E–I). Legs long, profemora stronger, protibiae nearly straight, weakly curved at apex inner side. Ventral surface of protarsomeres I–IV with a hairy brush. Inner side of mesotibiae weakly curved, extension at apex; ventral surface of mesotarsomeres I–IV with hairy brush. Ratio of length (width) of pro-, meso-, and metatibiae 23.0(4.0):28.0(4.0):38.0(4.0); that of metatarsomeres I–Ⅳ 10.0(4.0):7.0(4.0):7.0(4.0):12.0(4.0).

**Male genitalia** (Figure 6 and Figure 7L,M). Aedeagus length 3.9–4.0 mm long and width 1.1–1.2 mm. Parameres length 1.1–1.2 mm long and width 1.0–1.1 mm, cone-shape, apex acuminate; parameres widest at base, and straight narrowing in dorsal view; parameres slightly curved at apex, narrowed almost in a straight line up to apex in lateral view.

**Sexual dimorphism** (Figure 8B,D). Female body length 20.8–21.1 mm, width 9.6–9.7 mm. Body wider than male. Head 1.3 times as wide as interocular distance, pronotum 1.3 times as wide as long, elytra 1.5 times as long as wide. Antennae shorter than male, antennomeres VIII-X spherical. Pronotum widest at base, lateral margins subparallel from base to 2/3, then straightly narrowing toward anterior angles arcuated. Elytra more convex, 1/3 backward extension. Inner side of the protibiae weakly curved, upper spur sharper apically, almost to end of protarsomere I. Ventral surface of pro- and mesotarsomeres with a hairy brush. 

**Etymology**. This species is named from the Latin adjective “*politus**, -a*, *um*”, in reference to its smooth elytral surface.

**Distribution**. China, Sichuan.

**Diagnosis. Adult.** The new species is morphologically very similar to *N. quadrata* Li & Ren **sp. n****.**, making it difficult to distinguish from the latter by the external morphological characters alone. The identification was made by the combination of the DNA sequencing data and the following male character states: (1) pronotum and elytra convex on disc (pronotum and elytra strongly convex in *N. quadrata*); (2) antennomer Ⅶ longer than Ⅴ one (antennomer Ⅶ shorter than Ⅴ one in *N. quadrata*).


***Nalepa xinlongensis* Li & Ren sp. n. (Figure 10A,B)**


**Type locality**. China, Sichuan Province, Xinglong County, Mari.

**Type specimens** (Adults). **Holotype:** ♂, with the following labels: “2016. VIII.14//Mari Township, Xinlong County, Garzê, Sichuan, China//Xiu-Min Li et al. //Museum of Hebei University”//“30°42.158′ N//100°11.130′ E//Elev. 4028 m//Museum of Hebei University”. **Paratypes:** 9♂9♀ [2♂1♀ in ethanol] (MHBU), same data as holotype; 8♂2♀ [1♀ in ethanol] (MHBU), labeled “2016. VIII.10//Jitang Township, Chagyab County, Xizang, China//Xiu-Min Li et al. //Museum of Hebei University]”//“30°42.403′ N//97°18.685′ E//Elev. 3554 m//Museum of Hebei University”.

**Description.** Body length 16.5–16.7mm, width 6.7–6.8 mm; black, weakly shiny, oval-oblong. 

**Head** (Figure 9A,B). Anterior margin of epistome emarginated. Lateral margins of epistome straight. Lateral margins of head with indistinct emargination between epistome and genae. Head widest at eye level. Mentum transverse, with elliptical lateral sides, coarsely punctate and slightly impressed in the middle of the anterior edge. Antennae long and slender, reaching pronotal base when directed backwards, antennomeres VIII-X spherical (Figure 9C). Ratio of length/width of antennomeres Ⅱ-Ⅺ7.0(10.0):33.0(10.0):14.0(9.0):14.0(9.0):14.0(9.0):14.0(9.0):12.0(11.0):12.0(11.0):12.0(11.0):13.0(10.0).

**Thorax** (Figure 9D). Transverse, 1.5 times as wide as head. Widest at the middle, lateral margins weakly broading toward the middle, nearly subparallel, then toward anterior angles arcuated, ratio of width at the anterior margin to the middle part and base 20:31:30, external rim complete. Disc weakly convex, smooth, surface with dense punctation. Anterior angles obtuse, posterior ones nearly rectangular. Prosternal process obliquely sloping behind fore coxae, distinctly projecting beyond the margin of prosternum.

**Abdomen.** Long ovoid, 1.6 times as long as wide, widest at the middle, weakly arcuated, 2.7 times as long and 1.4 times as wide as pronotum, 2.2 times as wide as head. Strongly convex on disc, elytral surface smooth, with sparse punctures and dense irregular short wrinkles, apex of elytra steeply sloping, obtuse. Abdomen without hair tuft/short yellow setae between 1st and 2nd abdominal ventrites, 1st—3rd with transverse/longitudinal wrinkles, abdominal ventrites 4th—5th with dense punctures and simple particles.

**Legs** (Figure 9E–I). Legs long, profemora stronger, protibiae nearly straight. Ventral surface of protarsomeres I-IV with a hairy brush. Inner side of mesotibiae weakly curved, extension at apex; ventral surface of mesotarsomeres I–IV with a hairy brush. Ratio of length (width) of pro-, meso- and metatibiae 31.0(4.0):40.0(4.0):45.0(4.0); that of metatarsomeres I-Ⅳ 12(4):6.0(4.0):7.0(4.0):14.0(4.0).

**Male genitalia** (Figure 9K–M). Aedeagus length 3.8–3.9 mm and width 0.8–0.9 mm. Parameres length 1.2–1.3 mm and width 0.6–0.7 mm; parameres widest and convex at base, cone-shaped, straightly narrowing from base to apex laterally in dorsal view, with apex acuminate; dorsal side nearly straight, slightly curved to ventral side apically in lateral view; parameres narrowed almost in a straight line up to apex in lateral view.

**Sexual dimorphism** (Figure 10B). Female body length 18.1–18.3 mm, width 8.1–8.2 mm. Body wider than male. Head 1.35 times as wide as interocular distance, pronotum 1.3 times as wide as long, elytra 1.4 times as long as wide. Antennae shorter than male, antennomeres VIII-X spherical. Pronotum widest in middle, lateral margins subparallel from base to middle, then narrowing toward anterior angles arcuated. Elytra more convex, 1/3 backward extension. Inner side of protibiae nearly straight, upper apical spur obtuse. Ventral surface of pro- and mesotarsomeres with a hairy brush. 

**Etymology**. This species is named from the type locality.

**Distribution**. China, Sichuan; Xizang.

**Diagnosis.** The new species is morphologically very similar to *N. quadrata* Li & Ren **sp. n.**, but can be distinguished from the latter by the following male character states: (1) parameres cone-shaped, straightly narrowing from basal to apex dorsally, ratio of length (width) is higher than *N. quadrata*; (2) antennomeres Ⅳ–Ⅶ long and cylindrical (antennomeres Ⅳ-Ⅶ short and cylindrical in *N. quadrata*); (3) antennomeres VIII-X long and spherical (antennomeres VIII-X short and spherical in *N. quadrata*).


***Nalepa yushuensis* Li & Ren sp. n. (Figure 12A,B)**


**Type locality**. China, Qinghai Province, Yushu County, Jyêgu. 

**Type specimens** (Adults). **Holotype**: ♂, with the following labels: “2019.VII.24//Jyêgu Township, Yushu County, Qinghai, China//Xing-Long Bai et al. //Museum of Hebei University”//“33°05.257′ N//96°48.324′ E//Elev. 4008 m//Museum of Hebei University”. **Paratypes**: 26♂12♀ [4♂in ethanol] (MHBU), labeled “2018.VIII.23//Chaiwei Township, Qamdo County, Xizang, China]//Xiu-Min Li et al.//Museum of Hebei University”//“31°29.523′ N//97°13.723′ E//Elev. 3403 m//Museum of Hebei University”; 17♂5♀ (MHBU), labeled “2012.VII.21//Shanglaxiu Township, Yushu County, Qinghai, China//Xing-Long Bai *et al*.//Museum of Hebei University”//“32°57.479′ N//96°06.856′ E//Elev. 4227 m//Museum of Hebei University”.

**Description.** Body length 15.4–16.5 mm, width 6.2–6.5 mm; black, oval-oblong, elongated. 

**Head** (Figure 11A,B). Anterior margin of epistome emarginated. Lateral margins of epistome straight. Lateral margins of head with indistinct emargination between epistome and genae. Head widest at eye level. Mentum transverse, with elliptical lateral sides. Coarsely punctate and slightly impressed in the middle of the anterior edge. Antennae reaching the base of pronotum when directed backwards, antennomeres VIII-X spherical (Figure 11C). Ratio of length/width of antennomeres Ⅱ-Ⅺ 7.0(8.0):20.0(10.0):11.0(9.0):11.0(9.0):11.0(9.0):13.0(9.0):10.0(10.0):10.0(10.0):10.0(10.0):15.0(10.0).

**Thorax** (Figure 11D). Long and wide nearly equal, square, 1.6 times as wide as head. Widest at the middle, lateral margins subparallel from base to 2/3, then narrowing toward anterior angles arcuated, ratio of width at anterior margin to base 35:55, external rim complete. Disc strongly convex, smooth, surface with dense and fine punctation. Anterior angles obtuse, posterior ones rectangular. 

**Abdomen.** Long ovoid, 1.6 times as long as wide, widest at the middle, weakly arcuated, 2.35 times as long and 1.25 times as wide as pronotum, 2.0 times as wide as head. Strongly convex on disc, elytral surface smooth, with sparse punctures and shallow wrinkles, apex of elytra steeply sloping, obtuse. Abdomen without hair tuft/short yellow setae between 1st and 2nd abdominal ventrites, 1st—3rd with transverse/longitudinal wrinkles, abdominal ventrites 4th—5th with dense punctures and simple particles.

**Legs** (Figure 11E–I). Legs long, profemora stronger, protibiae nearly straight. Mesotibiae straight, extension at apex; ventral surface of pro, meso- and metatarsomeres I–IV with a hairy brush. Ratio of length (width) of pro-, meso- and metatibiae 29.0(4.4):30.0(4.4):36.0(5.0), that of metatarsomeres I–Ⅳ 10.0(3.0):5.0(3.0):5.0(3.0):10.0(3.0).

**Male genitalia** (Figure 11K–M). Aedeagus length 4.5–4.8 mm, width 1.1–1.2 mm. Parameres length 1.4–1.5 mm and width 0.8–0.9 mm; parameres wide and convex at base, apex acuminate, lateral margin arcuately narrowing from base to 1/3, straight narrowing from basal 1/3 to apex in dorsal view; parameres narrowed almost in a straight line up to apex, slightly curved, narrowed almost in a straight line up to apex in lateral view.

**Sexual dimorphism** (Figure 12B). Body length 17.5–18.0 mm, width 7.5–8.0 mm. Body wider than male. Head 1.3 times as wide as interocular distance, pronotum 1.2 times as wide as long, elytra 1.5 times as long as wide. Antennae shorter, not reaching base of pronotum when directed backwards, antennomeres VIII-X spherical. Pronotum and elytra more convex. Inner side of protibiae nearly staight, upper apical spur obtuse. 

**Etymology**. This species is named from the type locality.

**Distribution**. China, Qinghai.

**Diagnosis.** The new species is morphologically very similar to *N. cylindracea* Reitter, 1887, but can be distinguished from the latter by the following male character states: parameres lateral margin arcuately narrowing from base to 1/3, straight narrowing from basal 1/3 to apex (parameres cone-shape, apex acuminate, straightly narrowing from basal to apex in *N. cylindracea*). 


***Nalepa acuminata* Li & Ren sp. n. (Figure 14A,B)**


**Type locality**. China, Sichuan Province, Baiyü County, Hepo. 

**Type specimens** (Adults). **Holotype**: ♂, with the following labels: “2016.VIII.6//Hepo Township, Baiyü County, Sichuan, China//Xiu-Min Li et al.//Museum of Hebei University”//“31°22.899′ N//98°53.359′ E//Elev. 3000 m//Museum of Hebei University”. **Paratypes**: 12♂11♀ [2♂1♀ in ethanol] (MHBU), same data as holotype; 1♂ [2♀ in ethanol] (MHBU), labeled “2016.VIII.6//Jinsha Township, Baiyü County, Sichuan, China//Xiu-Min Li et al.//Museum of Hebei University”//“31°16.522′ N//98°48.017′ E//Elev. 2980 m//Museum of Hebei University”; 1♀ [1♂ in ethanol] (MHBU), labeled “2016.VIII.6//Xindu Township, Luhuo County, Sichuan, China//Xiu-Min Li et al.//Museum of Hebei University”//“31°23.176′ N//100°41.359′ E//Elev. 3194 m//Museum of Hebei University”.

**Description.** Body length 18.1–18.5 mm, width 7.5–7.6 mm; black, weakly shiny, oval-oblong. 

**Head** (Figure 13A,B). Clypeus nearly straight, anterior margin of frontoclypeus weakly bisinuate, covered with dense punctation. Head widest at eye level. Outer margins of head between frontoclypeus and genae with obtuse distinct emargination. Mentum transverse, with elliptical lateral sides. coarsely punctate and slightly impressed in the middle of the anterior edge. Antennae reaching the base of pronotum when directed backwards, antennomeres 8–10 spherical (Figure 13G). Ratio of length/width of antennomeresⅡ-Ⅺ 5.0(8.0):27.0(8.0):13.0(8.0):13.0(7.0):12.0(7.0):16.0(8.0):10.0(10.0):10.0(10.0):10.0(10.0):14.0(10.0).

**Thorax** (Figure 13H). Transverse, nearly trapezoid-shaped, 1.8 times as wide as head. Widest at base, lateral margins weakly narrowing toward the middle, then toward anterior angles arcuated, ratio of width at anterior margin to the middle part and base 16:25:27, external rim complete. Disc weakly convex, smooth, surface with dense punctation. Anterior angles obtuse, posterior ones nearly rectangular. Prosternal process obliquely sloping behind fore coxae, distinctly projecting beyond margin of prosternum.

**Abdomen.** Long ovoid, 1.5 times as long as wide, widest at the middle, weakly arcuated, 2.3 times as long and 1.5 times as wide as pronotum, 2.1 times as wide as head. Strongly convex on disc, elytral surface smooth, with sparse punctures and dense irregular short wrinkles, apex of elytra steeply sloping, obtuse. Abdomen with hair tuft/short yellow setae between 1st and 2nd abdominal ventrites, 1st–3rd with transverse/longitudinal wrinkles, abdominal ventrites 4th–5th with dense punctures and simple particles.

**Legs** (Figure 13C–F). Legs long, profemora stronger, protibiae weakly curved. Ventral surface of protarsomeres 1–4 with a hairy brush. Inner side of mesotibiae weakly curved, extension at apex; ventral surface of mesotarsomeres 1–4 with hairy brush. Ratio of length (width) of pro-, meso- and metatibiae 33.0(4.0):35.0(4.0):57.0(4.0), that of metatarsomeres 1–4 16.0(4.0):10.0(4.0):8.0(4.0):15.0(4.0).

**Male genitalia** (Figure 13J–L). Aedeagus length 4.8–5.0 mm, width 1.1–1.2 mm. Parameres length 1.5–1.6 mm long, width 0.9–1.0 mm; parameres wide and convex at base, bottleneck-shaped, arcuately narrowing from base to basal 1/3 in dorsal view, straight narrowing at basal 1/3, then to apex acuminate; slightly curved to ventral side apically in lateral view; parameres slightly curved, narrowed almost in a straight line up to apex in lateral view.

**Sexual dimorphism** (Figure 14B). Body length 16.9–17.1 mm, width 8.0–8.1 mm. Body wider than male. Head 1.3 times as wide as interocular distance, pronotum 1.4 times as wide as long, elytra 1.4 times as long as wide. Antennae shorter, antennomeres 8–10 spherical. Pronotum widest at base, lateral margins subparallel from base to middle, then narrowing toward anterior angles arcuated. Elytra more convex. Inner side of protibiae weakly curved, upper apical spur obtuse. 

**Etymology**. This species is named from the Latin adjective “*acuminatus*, *-a*, *um*”, in reference to its bottle-neck shape in the end of aedoeagus.

**Distribution**. China, Sichuan.

**Diagnosis**. The new species is easy to distinguish from the other species of *Nalepa* by the following male character states: (1) pronotum nearly trapezoid-shaped, widest at base, lateral margins narrowing toward middle; (2) aedeagus bottleneck-shaped, arcuately narrowing from base to basal 1/3 in dorsal view, then straight narrowing at basal 1/3.


***Nalepa ovalifolia* Li & Ren sp. n. (Figure 16A,B)**


**Type locality**. China, Xizang, Jomda County, Gyamda. 

**Type specimens** (Adults). **Holotype**: ♂, with the following labels: “2016.VIII.8//Gyamda Township, Jomda County, Xizang, China//Xiu-Min Li et al. //Museum of Hebei University”//“31°27.610′ N//98°10.722′ E//Elev. 3630 m//Museum of Hebei University”. **Paratypes**: 6♂4♀ [2♂ in ethanol] (MHBU), same data as holotype; 20♂19♀ [2♂2♀ in ethanol] (MHBU), labeled “2016.VIII.8//Kargang Township, Jomda County, Xizang, China//Xiu-Min Li et al.//Museum of Hebei University”//“31°26.669′ N//98°09.032′ E//Elev. 3700 m//Museum of Hebei University]”; 1♂1♀ (MHBU), labeled “2016.VIII.7//Tangpu Township, Jomda County, Xizang, China//Xiu-Min Li et al. //Museum of Hebei University”//“31°35.091′ N//98°21.427′ E//Elev. 3293 m//Museum of Hebei University”; 8♂6♀ [1♀ in ethanol] (MHBU), labeled “2016.VIII.9//Yulong Township, Jomda County, Xizang, China//Xiu-Min Li et al.//Museum of Hebei University”//“31°22.014′ N//97°48.436′ E//Elev. 4164 m//Museum of Hebei University”; 6♂[1♂1♀ in ethanol] (MHBU), labeled “2016.VIII.5//Ronggai Township, Baiyü County, Sichuan, China//Xiu-Min Li et al. //Museum of Hebei University”//“31°07.621′ N//98°55.026′ E//Elev. 3120 m//Museum of Hebei University]”; 3♂1♀ [3♂2♀ in ethanol] (MHBU), labeled “2016.VIII.6//Jangra Township, Dêgê County, Garzê Zhou [similar to City], Sichuan, China//Xing-Long Bai et al. //Museum of Hebei University”//“31°41.912′ N//98°34.786′ E//Elev. 3120 m//Museum of Hebei University”.

**Description.** Body length 21.0–21.5 mm, width 8.0–8.2 mm; black, oval-oblong, elongated. 

**Head** (Figure 15A,B). Anterior margin of epistome emarginated. Lateral margins of epistome straight. Lateral margins of head with indistinct emargination between epistome and genae. Head widest at eye level. Mentum transverse, with elliptical lateral sides. Coarsely punctate and slightly impressed in the middle of anterior edge. Antennae reaching base of pronotum when directed backwards, antennomeres VIII-X spherical (Figure 15C). Ratio of length/width of antennomeres Ⅱ-Ⅺ 8.0(8.0):30.0(8.0):13.0(8.0):14.0(8.0):14.0(8.0):14.0(8.0):10.0(10.0):10.0(10.0):10.0(10.0):13.0(10.0).

**Thorax** (Figure 15D). Long and wide nearly equal, square, 1.6 times as wide as head. Widest at the base, lateral margins subparallel from base to 2/3, then narrowing toward anterior angles arcuated, ratio of width at anterior margin to base 26:42, external rim complete. Disc strongly convex, smooth, surface with dense and fine punctation. Anterior angles obtuse, posterior ones rectangular. Prosternal process obliquely sloping behind procoxae, distinctly projecting beyond margin of prosternum.

**Abdomen.** Long ovoid, 1.7 times as long as wide, widest at the middle, weakly arcuated, 2.4 times as long and 1.25 times as wide as pronotum, 2.0 times as wide as head. Strongly convex on disc, elytral surface smooth, with sparse punctures and shallow wrinkles, apex of elytra steeply sloping, obtuse. Abdomen without hair tuft/short yellow setae between 1st and 2nd abdominal ventrites, 1st–3rd with transverse/longitudinal wrinkles, abdominal ventrites 4th—5th with dense punctures and simple particles.

**Legs** (Figure 15E–I). Legs long, profemora stronger, protibiae nearly straight. Ventral surface of protarsomeres I–IV with a hairy brush. Mesotibiae straight, extension at apex; ventral surface of mesotarsomeres I–IV with a hairy brush. Ratio of length (width) of pro-, meso- and metatibiae 35.0(4.4):36.0(4.4):50.0(5.0), that of metatarsomeres I–Ⅳ 10.0(3.0):5.0(3.0):5.0(3.0):10.0(3.0).

**Male genitalia** (Figure 15K–M). Aedeagus length 4.0–4.2 mm and width 0.9–1.0 mm. Parameres length 1.4–1.5 mm and width 0.7–0.8 mm; parameres wide and convex at base, cone-shaped, straightly narrowing from base to apex laterally in dorsal view, with apex acuminate; dorsal side nearly straight, slightly curved to ventral side apically in lateral view; parameres narrowed almost in a straight line up to apex in lateral view.

**Sexual dimorphism** (Figure 16B). Body length 21.0–21.5 mm, width 9.0–9.4 mm. Body wider than male. Head 1.3 times as wide as interocular distance, pronotum 1.2 times as wide as long, elytra 1.5 times as long as wide. Antennae shorter, antennomeres VIII-X oval-shape. Pronotum and elytra more convex. Inner side of protibiae weakly curved, upper apical spur obtuse. 

**Etymology**. This species is named from the Latin adjective “*ovalifolius*, *a*, *um*”, in reference to its ovalis elytral end. 

**Distribution**. China, Xizang.

**Diagnosis.** The new species is morphologically very similar to *N. acuminata*
**sp. n.**, but can be distinguished from the latter by the following male character states: (1) male body more elongated than *N.*
*acuminata*; (2) elytra long ovoid (elytra more oval in *N. acuminata*). 

### 3.2. Phylogenetic Relationships and Species Delimitation

The final, concatenated dataset was 3571 bp long (COI, 744 bp and 648bp; COⅡ, 661 bp; Cytb, 579 bp; 16S, 496 bp; 28S-D2, 443 bp), including 144 sequences from 13 specimens of the type species and 16 specimens of the six new species: *N.*
*acuminata*
**sp. n.** (2 specimens), *N. yushuensis*
**sp. n.** (3 specimens)**,**
*N. xinlongensis*
**sp. n.** (1 specimens), *N. ovalifolia*
**sp. n.** (3 specimens), *N.*
*polita*
**sp. n****.** (3 specimens), *N. quadrata*
**sp. n.** (4 specimens) (Appendix A).

The ML tree revealed that the larval sample TiBN02 and the adults of *N. cylindracea* belong to a clade, and the larval sample SCN04 and the adults of *N.*
*quadratus* sp. n. belong to a clade. Further five molecular species delimitation analyses confirmed that TiBN02 is the larva of *Nalepa cylindracea*, and SCN04 is the larva of *Nalepa*
*quadratus*
**sp. n.** Meanwhile, all sampled individuals were grouped into seven clades (corresponding to seven species), and the interspecific relationships were well supported overall (6 nodes with uBV > 95%). The use of ABGD with COI, COⅡ, and Cytb yielded very similar results except *N.*
*polita*
**sp. n****.** on Cytb. Overall, the species delimitation methods gave comparable results and were mostly congruent. The main discrepancies were found with GMYC and PTP, where the analyses delivered overestimated the number of MOTUs results compared to ABGD methods about three species: *N. cylindracea*, *N. ovalifolia*
**sp. n.,** and *N.*
*polita*
**sp. n****.** (Figure 17).

### 3.3. Geographical Distribution and Bionomics

All species of the genus *Nalepa* are found to exhibit characteristic distribution patterns within a small geographical range, and are only distributed in southern Qinghai, northwestern Sichuan, and eastern Xizang, China (Figure 18). 

Interestingly, a vertical gradient of species diversity was observed in the *Nalepa* species, with the highest species richness found at an elevation between 3000–5000 m, suggesting an adaptation of the *Nalepa* species to semi-arid and arid environments. In the field, the larvae and adults were usually found under big stones or shelters, probably feeding on decaying plant roots or leaves (Figure 19). 

## 4. Discussion

### 4.1. Species Delimitation

Different molecular species delimitation analyses resulted in 7 to 12 putative species, usually more than the seven morphospecies recognized here. This is consistent with the findings that both GMYC and PTP have a tendency to overestimate the number of MOTUs [27,28,29]. The three specimens of *Nalepa*
*polita*
**sp. n.** are not easily distinguished because they are very similar in morphological characteristics (Figure 6, Figure 7 and Figure 8), although they are defined as different species by most molecular species delimitation analyses except ABGD on Cytb. Here we still consider them as a species, and these results may be related to the short speciation time, but it is worth further investigation with more specimens from a wider distribution area. The diversity patterns of insects tend to concentrate in many mountainous areas providing a variety of habitats for flora and fauna [28]. Interestingly, *Nalepa* diversity of species exhibit at high elevation mountains. So, the mountains of the northeastern Tibetan Plateau probably act as rapid evolutionary cradles for *Nalepa* diversity. 

Overall, the taxonomic status of six new species was confirmed by molecular species delimitation analyses and morphological evidence: *N.*
*acuminata*
**sp. n.**, *N. yushuensis*
**sp. n.,**
*N. xinlongensis*
**sp. n.,**
*N. ovalifolia*
**sp. n.**, *N.*
*polita*
**sp. n****.**, and *N. quadrata*
**sp. n.** (Figure 17).

### 4.2. Phylogeny and Systematic Status of the Genus Nalepa

The genus *Nalepa* was erected 1887, without any update on the new species record before this study. Here, we described six new species, larvae of *N. cylindracea* Reitter, 1887, and *N. quadrata* Li & Ren, **sp. n.** Our study indicated that all species of this genus are mainly distributed in 3000–5000 m, and the distribution of seven species showed a significant bias towards mountains. Mountains and high-altitude environments are strong drivers of adaptive evolution in endemic species [28,30]. The Tibetan Plateau is widely considered a large natural experimental area for speciation, with numerous studies suggesting that its uplift has facilitated allopatric speciation and diversification since the Miocene [31,32,33,34,35,36,37]. *Naepa* is an endemic genus from the Tibetan Plateau, where they have restricted areas of distribution. The uplifting events of the Tibetan Plateau restricted gene flow between species or populations, and likely influenced species diversity formation of *Nalepa*. Different groups or organisms can undergo convergent evolution during adaptation to similar habitats in the Tibetan Plateau [30]. *Nalepa* was initially considered a subgenus of the genus *Blaps*, and then was elevated. Although *Nalepa* is currently treated as a valid genus, its monophyly and its phylogenetic placement within Blaptini still need to be verified, especially with molecular data. 

Although the classification of Blaptini is well developed, the rank of some taxa and their position in the tribe remains unclear due to lacking enough specimens. The phylogenetic relationships within this tribe are not well supported by molecular evidence to date. In this study, we collected DNA sequences of seven *Nalepa* species, which will provide valuable data on the genus for further studies on the molecular phylogeny of the tribe Blaptini.

## 5. Conclusions

In the present study, six new species and the larvae of two of them were described and illustrated from the northeast region of the Tibetan Plateau (Xizang, Sichuan and Qinghai, China). Phylogenetic relationships of 29 specimens of the genus *Nalepa* are provided and discussed based on fragments of four mitochondrial markers (cytochrome coxidase subunit I, COI; cytochrome coxidase subunit Ⅱ, COⅡ; cytochrome b, Cytb; 16S ribosomal DNA, 16S), and one nuclear marker (28S ribosomal DNA domain D2, 28S-D2) molecular species delimitation methods to the taxonomic status of the new species and larvae. 

Lastly, the status of the genus *Nalepa* from the Tibetan Plateau is briefly discussed. *Nalepa* was initially considered as one subgenus of the genus *Blaps*, and then was elevated from subgenus to genus. Although *Nalepa* is currently treated as a valid genus, its monophyly and its phylogenetic placement within Blaptini still need to be verified, especially with molecular data. 

## Figures and Tables

**Figure 1 insects-13-00598-f001:**
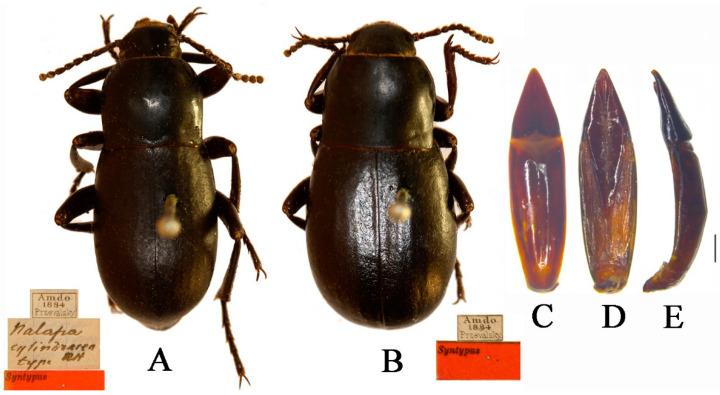
*Nalepa cylindracea.* (**A**,**B**) Habitus: (**A**) male; (**B**) female. (**C**–**E**) aedeagus: (**C**) dorsal view; (**D**) ventral view; (**E**) lateral view. Scale bars: 1.0 mm.

**Figure 2 insects-13-00598-f002:**
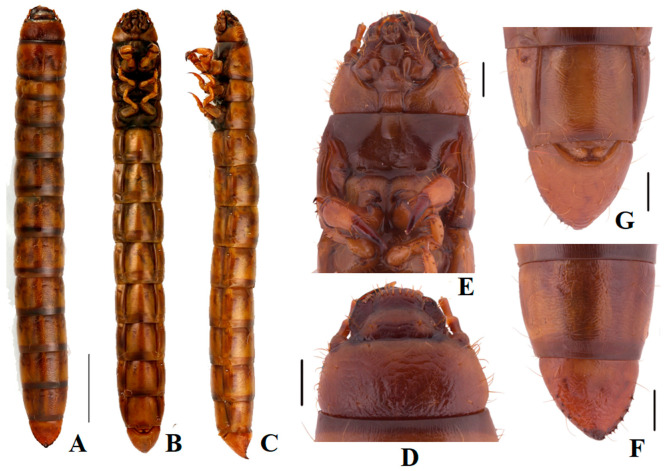
Larva of *Nalepa cylindracea*. (**A**–**C**) Habitus: (**A**) dorsal view; (**B**) ventral view; (**C**) lateral view. (**D**) Head, dorsal view; (**E**) Head and prothoracic leg, ventral view; (**F**) Pygopods, dorsal view; (**G**) Pygopods, ventral view. Scale bars: 5 mm (**A**–**C**), 1 mm (**D**–**G**).

**Figure 3 insects-13-00598-f003:**
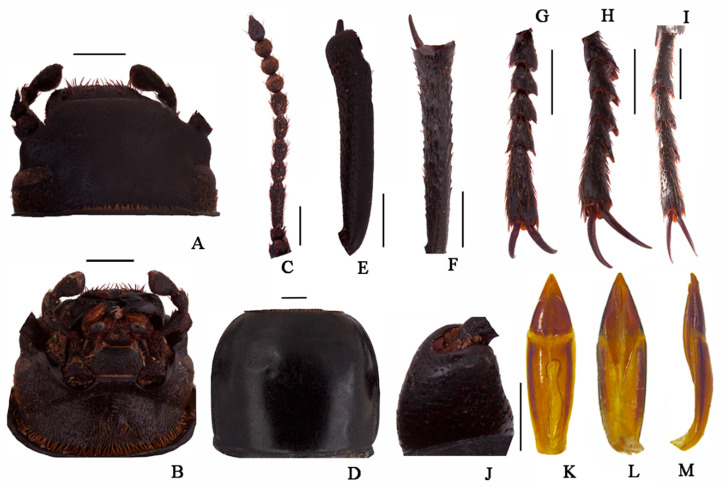
*Nalepa quadrata* Li & Ren, **sp. n.** (**A**) head, dorsal view; (**B**) head, ventral view; (**C**) antenna; (**D**) pronotum; (**E**) protibia; (**F**) mesotibia; (**G**) protarsus; (**H**) mesotarsus; (**I**) metatarsus; (**J**) profemora; (**K**–**M**) aedeagus: (**K**) dorsal view; (**L**) ventral view; (**M**) lateral view. Scale bars: (**A**–**J**) (1.0 mm), (**K**–**M**) (1.0 mm).

**Figure 4 insects-13-00598-f004:**
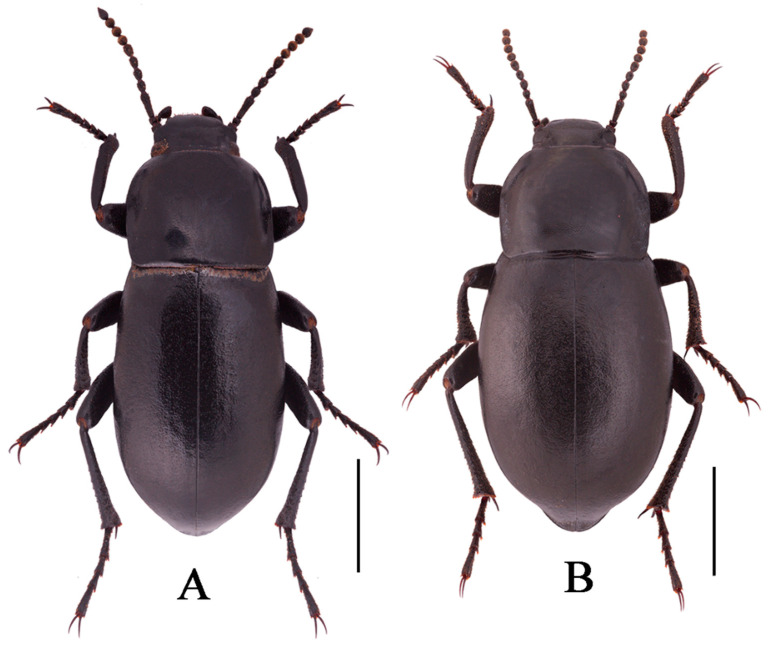
Habitus of *N. quadrata* Li & Ren, **sp. n.** ((**A**) male, holotype; (**B**) female, paratype). Scale bars: 5.0 mm.

**Figure 5 insects-13-00598-f005:**
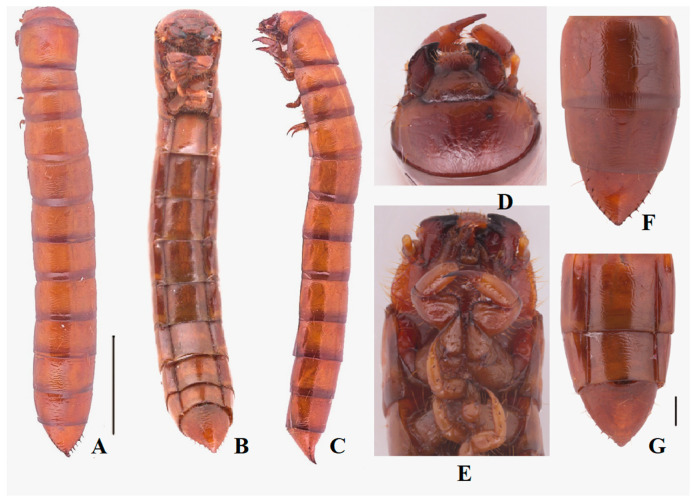
Larva of *Nalepa quadrata* Li & Ren **sp. n.** (**A**–**C**) Habitus: (**A**) dorsal view; (**B**) ventral view; (**C**) lateral view. (**D**) Head, dorsal view; (**E**) Head and prothoracic leg, ventral view; (**F**) Pygopods, dorsal view; (**G**) Pygopods, ventral view. Scale bars: 5 mm (**A**–**C**), 1 mm (**D**–**G**).

**Figure 6 insects-13-00598-f006:**
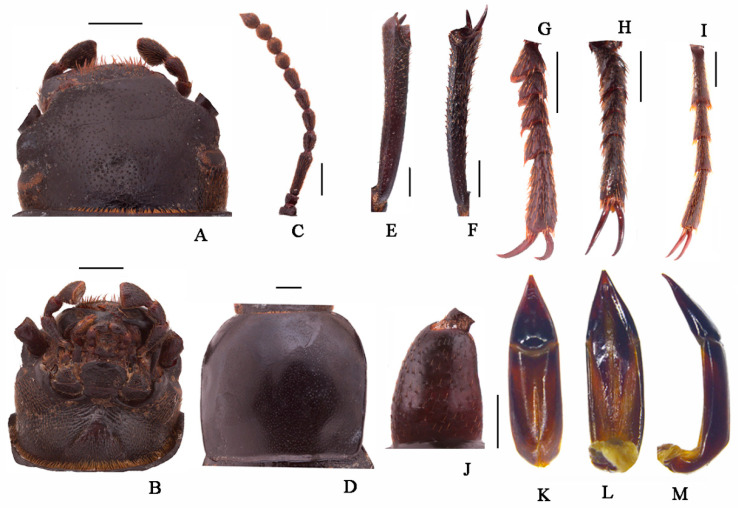
*Nalepa polita* Li & Ren **sp.**
**n.** (**A**) head, dorsal view; (**B**) head, ventral view; (**C**) antenna; (**D**) pronotum; (**E**) protibia; (**F**) mesotibia; (**G**) protarsus; (**H**) mesotarsus; (**I**) metatarsus; (**J**) profemora; (**K**–**M**) aedeagus: (**K**) dorsal view; (**L**) ventral view; (**M**) lateral view. Scale bars: (**A**–**J**) (1.0 mm), (**K**–**M**) (1.0 mm); from Chowa, Dêgê, Sichuan, China.

**Figure 7 insects-13-00598-f007:**
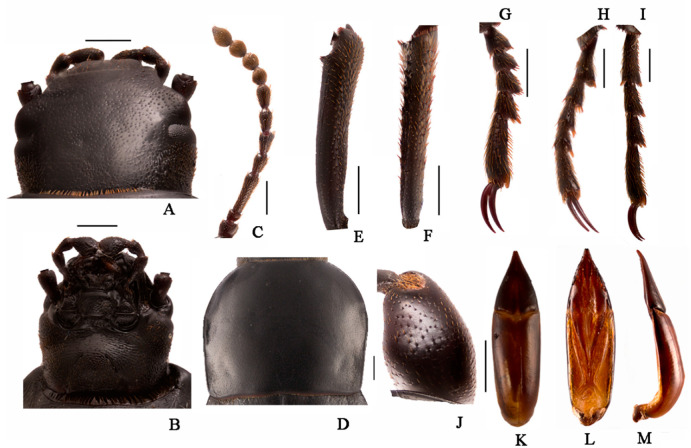
*Nalepa polita* Li & Ren **sp.**
**n.** (**A**) head, dorsal view; (**B**) head, ventral view; (**C**) antenna; (**D**) pronotum; (**E**) protibia; (**F**) mesotibia; (**G**) protarsus; (**H**) mesotarsus; (**I**) metatarsus; (**J**) profemora; (**K**–**M**) aedeagus: (**K**) dorsal view; (**L**) ventral view; (**M**) lateral view. Scale bars: (**A**–**J**) (1.0 mm), (**K**–**M**) (1.0 mm); from Gyawa, Garzê, Sichuan, China.

**Figure 8 insects-13-00598-f008:**
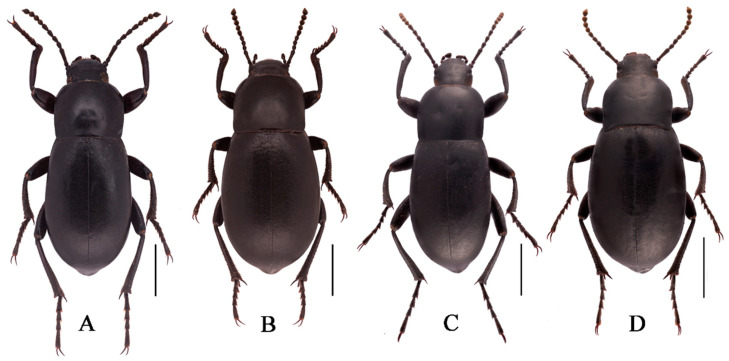
Habitus of *N. polita*
**sp. n.** (**A**) male, holotype; (**B**) female, paratype; from Chowa, Dêgê, Sichuan, China; (**C**,**D**) male, female; from Gyawa, Garzê, Sichuan, China. Scale bars: 5.0 mm.

**Figure 9 insects-13-00598-f009:**
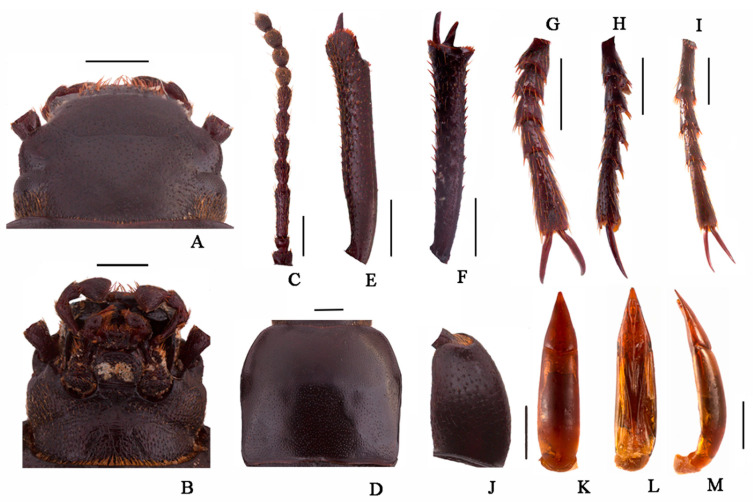
*Nalepa xinlongensis***sp. n.** (**A**) head, dorsal view; (**B**) head, ventral view; (**C**) antenna; (**D**) pronotum; (**E**) protibia; (**F**) mesotibia; (**G**) protarsus; (**H**) mesotarsus; (**I**) metatarsus; (**J**) profemora; (**K**–**M**) aedeagus: (**K**) dorsal view; (**L**) ventral view; (**M**) lateral view. Scale bars: 1.0 mm.

**Figure 10 insects-13-00598-f010:**
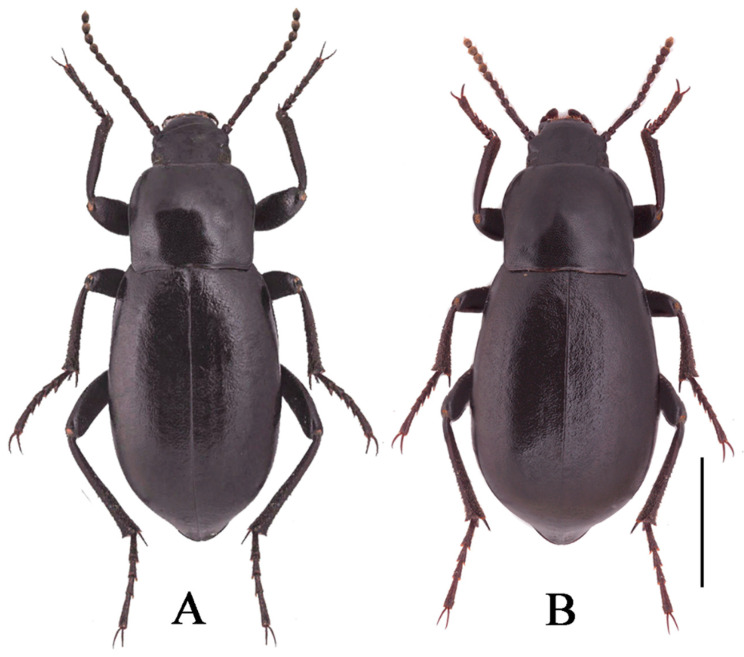
Habitus of *N. xinlongensis*
**sp. n.** ((**A**) male, holotype; (**B**) female, paratype). Scale bars: 5.0 mm.

**Figure 11 insects-13-00598-f011:**
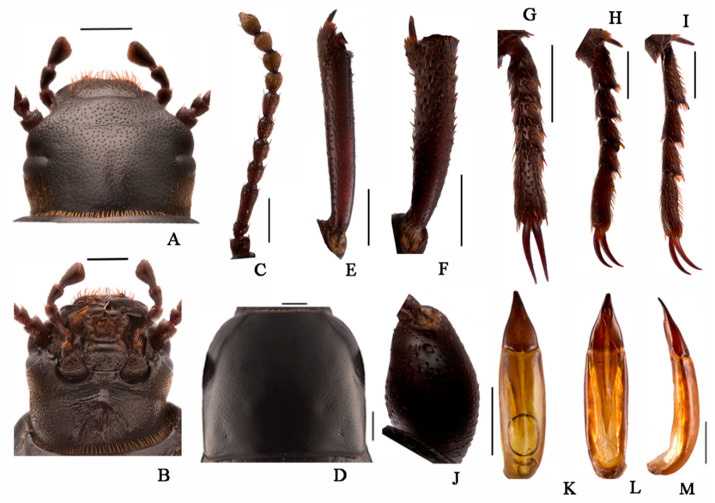
*Nalepa yushuensis***sp. n.** (**A**) head, dorsal view; (**B**) head, ventral view; (**C**) antenna; (**D**) pronotum; (**E**) protibia; (**F**) mesotibia; (**G**) protarsus; (**H**) mesotarsus; (**I**) metatarsus; (**J**) profemora; (**K**–**M**) aedeagus: (**K**) dorsal view; (**L**) ventral view; (**M**) lateral view. Scale bars: 1.0 mm.

**Figure 12 insects-13-00598-f012:**
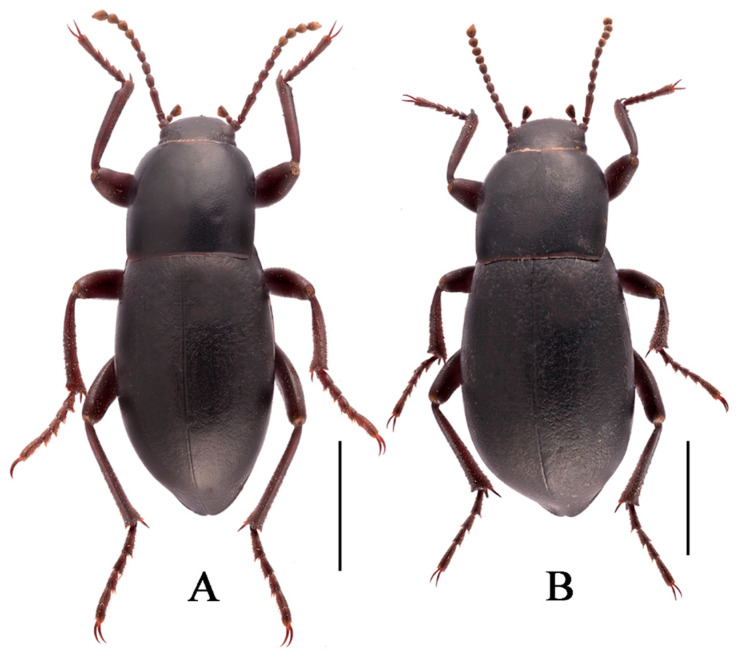
Habitus of *N. yushuensis*
**sp. n.** ((**A**) male, holotype; (**B**) female, paratype). Scale bars: 5.0 mm.

**Figure 13 insects-13-00598-f013:**
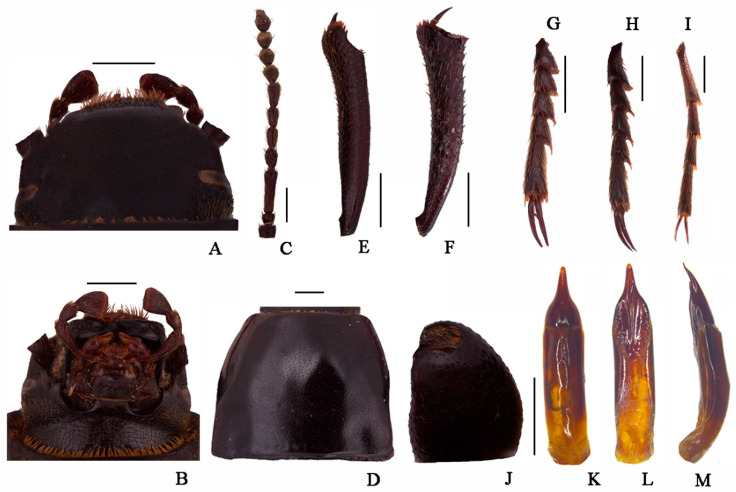
*Nalepa acuminata* Li & Ren **sp. n.** (**A**) head, dorsal view; (**B**) head, ventral view; (**C**) antenna; (**D**) pronotum; (**E**) protibia; (**F**) mesotibia; (**G**) protarsus; (**H**) mesotarsus; (**I**) metatarsus; (**J**) profemora; (**K**–**M**) aedeagus: (**K**) dorsal view; (**L**) ventral view; (**M**) lateral view. Scale bars: 1.0 mm.

**Figure 14 insects-13-00598-f014:**
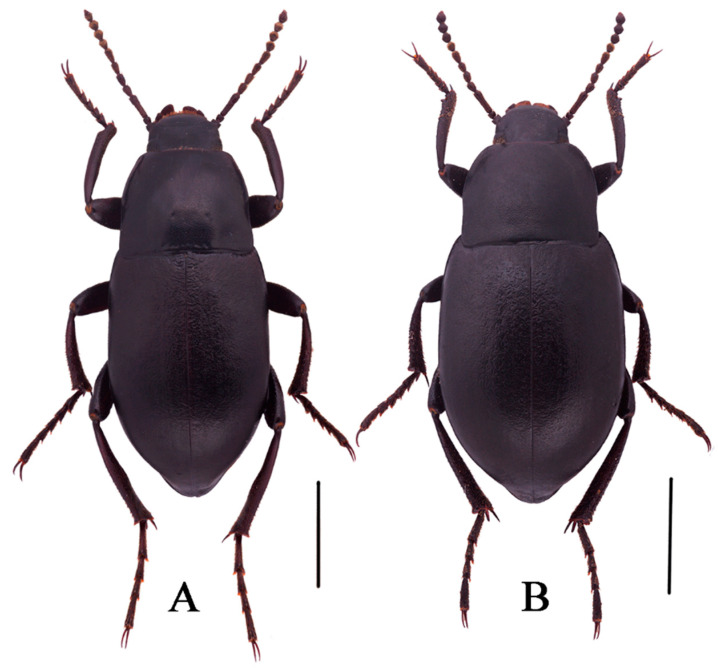
Habitus of *N. acuminata*
**sp. n.** ((**A**) male, holotype; (**B**) female, paratype). Scale bars: 5.0 mm.

**Figure 15 insects-13-00598-f015:**
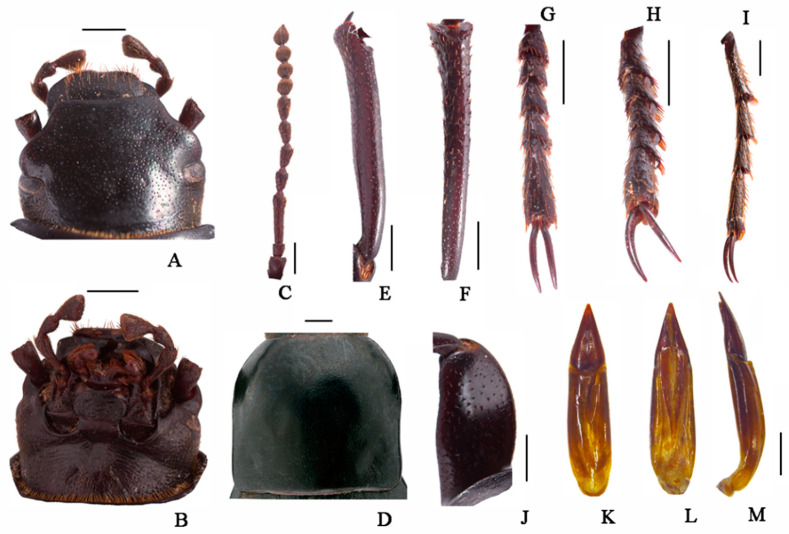
*Nalepa ovalifolia***sp. n.** (**A**) head, dorsal view; (**B**) head, ventral view; (**C**) antenna; (**D**) pronotum; (**E**) protibia; (**F**) mesotibia; (**G**) protarsus; (**H**) mesotarsus; (**I**) metatarsus; (**J**) profemora; (**K**–**M**) aedeagus: (**K**) dorsal view; (**L**) ventral view; (**M**) lateral view. Scale bars: 1.0 mm.

**Figure 16 insects-13-00598-f016:**
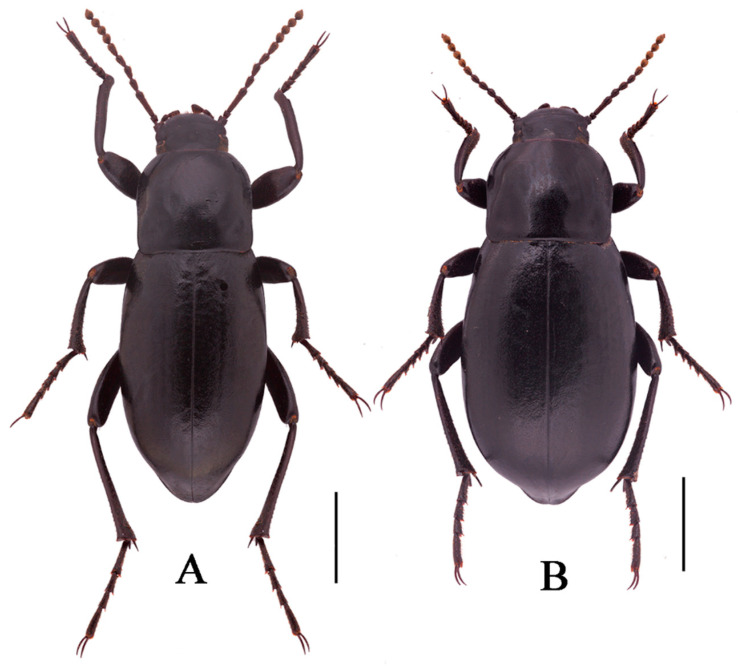
Habitus of *N. ovalifolia*
**sp. n.** ((**A**) male, holotype; (**B**) female, paratype). Scale bars: 5.0 mm.

**Figure 17 insects-13-00598-f017:**
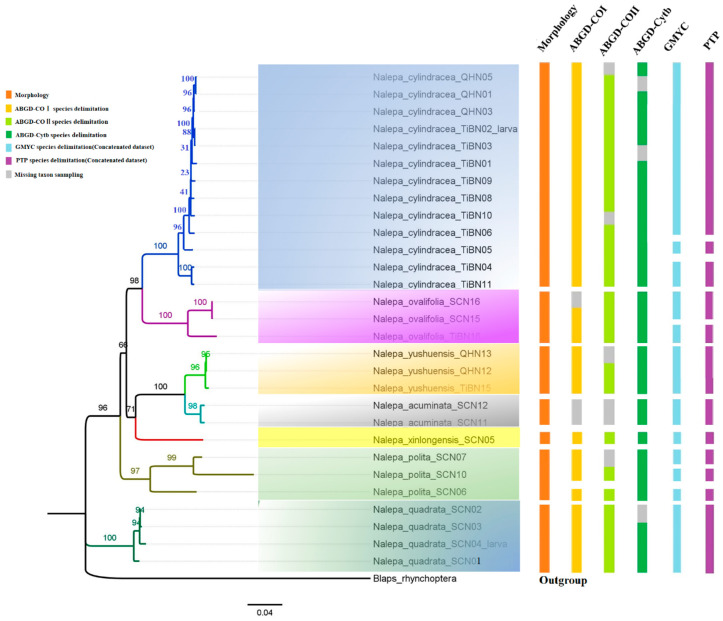
Maximum likelihood phylogenetic tree based on 3571 bp of mitochondrial and nuclear DNA sequences (*COI*, *COⅡ*, *Cytb*, *16S* rDNA, and *28S* rDNA) within the genus *Nalepa*. Support for each node is represented by ultrafast bootstrap values (uBV).

**Figure 18 insects-13-00598-f018:**
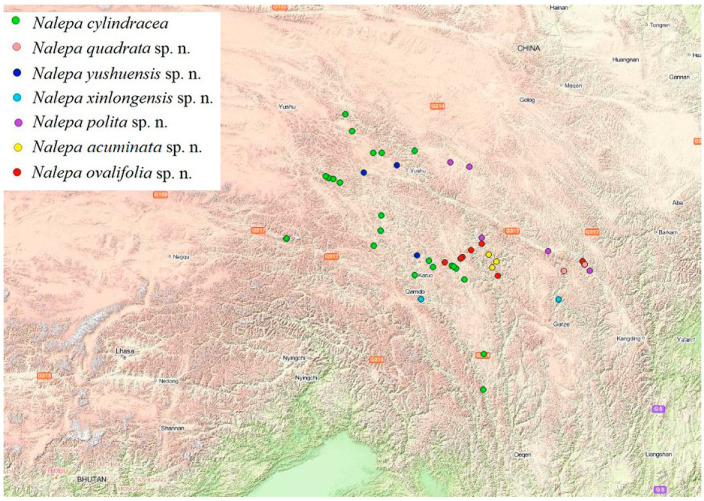
The geographical distribution of 49 samples of *Nalepa* used in this study. Each represents a separate species.

**Figure 19 insects-13-00598-f019:**
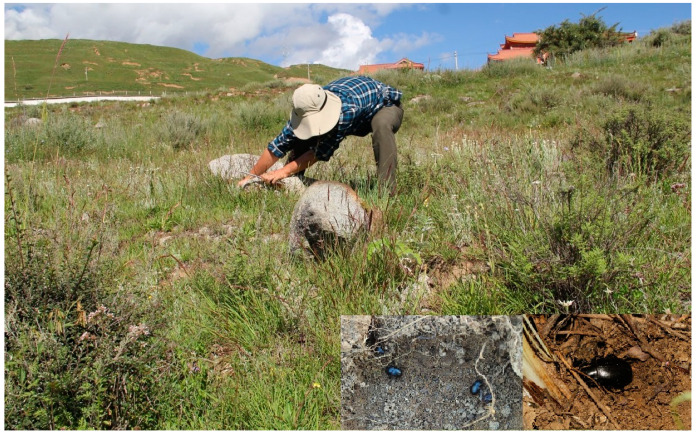
Habitat for *Nalepa*
*polita* sp. n. photo by Xiumin Li, at Rinda, Luhuo, Sichuan, China, on 3 August 2016.

**Table 1 insects-13-00598-t001:** Primer sequences for PCR.

Gene	Primer (Forward/Reverse/)	Sequence (Forward and Reverse) 5′→3′	PCR Conditions (Annealing)	References
COI	F 2183	CAACATTTATTTTGATTTTTTGG	50 °C	Monteiro & Pierce, 2001 [18]
	R 3014	TCCAATGCACTAATCTGCCATATTA		
COI	F 1490	GGT CAA ATC ATA AAG ATA TTG G	50 °C	Simon et al.,
	R 2198	TAA ACT TCA GGG TGA CCA AAA AAT CA		1994 [19]
COII	F 3037	ATG GCA GAT TAG TGC AAT GG	50 °C	Simon et al.,
	R 3785	GTT TAA GAG ACC AGT ACT TG		1994 [19]
Cytb	F revcb2h	TGAGGACAAATATCATTTTGAGGW	50 °C	Simmons et al., 2001 [20]
	R rebcbj	TCAGGTCGAGCTCCAATTCATGT		
16S	F 13398	CGCCTGTTTATCAAAAACAT	50 °C	Simon et al., 1994 [19]
	R 12887	CCGGTCTGAACTCAGATCAT		
28S-D2	F 3665	AGAGAGAGTTCAAGAGTACGTG	58 °C	Belshaw et al., 1997 [7]
	R 4068	TTGGTCCGTGTTTCAAGACGGG		

## Data Availability

The data of the research were deposited at the College of Life Sciences, Hebei University, Baoding, China. The specimens associated with this paper, as well as the sequence data submitted to GenBank, conform to the 2014 Nagoya Protocol on Access to Genetic Resources and the Fair and Equitable Sharing of Benefits Arising from their Utilization (https://www.cbd.int/abs/, accessed on 26 June 2022).

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
