# Peer review of "Systematic Review of the Genus Nalepa Reitter, 1887 (Coleoptera, Tenebrionidae, Blaptinae, Blaptini) from the Tibetan Plateau, with Description of Six New Species and Two Larvae [Author-notes fn1-insects-13-00598]"

_insects, 2022, doi:10.3390/insects13070598_

Round 1

Reviewer 1 Report

The manuscript needs some improvements in Material and Methoda, text formatting, supporting values on the phylogenetic tree are invisible, key to species should be rewritten. Male genitalia are to small, they are one of the most importnat character to separate new species but tey are simply too small to see any details. Re-desription of the type species is missing, it is hard to compare newly described species with the type species. In the Results different approaches used to delimit OTUs (later new species) are not discussed, and each shows slighty different partitioning. I recomend minor revision of that manuscript. Some of my suggestions are made in the attached pdf.

Reviewer 2 Report

Dear authors, 

please find all my suggestions embodied in your manuscript. Consider to revise carefully the organisation of your text, as well as the discussion section. Anyhow, congratulation for your very interesting findings!

Please do not hesitate to contact me for additional information.

Best regards, 

MR
